# Anisotropic Scaling Non-Relativistic Holography: A Symmetry Perspective

**Hong Lü** [1,2,3]**, Pujian Mao** [1,]*** and Junbao Wu** [1,2]

1 Center for Joint Quantum Studies and Department of Physics, School of Science, Tianjin University, 135 Yaguan Road, Tianjin 300350, China
2 Peng Huanwu Center for Fundamental Theory, Hefei 230026, China
3 Joint School of National University of Singapore and Tianjin University, International Campus of Tianjin University, Binhai New City, Fuzhou 350207, China
* Correspondence: pjmao@tju.edu.cn

**Abstract:** We study the holographic dual of the two-dimensional non-relativistic field theory with anisotropic scaling from a symmetry perspective. We construct a new four-dimensional metric with two-dimensional global anisotropic scaling isometry. The four-dimensional spacetime is homogeneous and is a solution of Einstein gravity with quadratic-curvature extension. We consider this spacetime dual to the vacuum of the boundary field theory. By introducing a proper solution phase space, we find that the asymptotic symmetry of the gravity theory is the two-dimensional local anisotropic conformal symmetry, which recovers precisely the results from the dual non-relativistic field theory side.

**Keywords:** non-relativistic holography; asymptotic symmetry; anisotropic scaling

## 1. Introduction

An interesting feature of quantum field theory (QFT) with scaling symmetry in two-dimensional (2D) spacetime is that the global symmetry can be enhanced to an infinite-dimensional local symmetry. The best known example is revealed by Polchinski in [1] that a local, unitary Poincare-invariant 2D QFT with a global scaling symmetry and a discrete non-negative spectrum of scaling dimensions must have both a left and a right local conformal symmetry. More than ten years ago, Hofman and Strominger showed that for a chiral situation, the local conformal symmetry is still implied [2], which leads to two kinds of minimal theories, namely the 2D conformal field theory (CFT) [3] or the 2D warped conformal field theory (WCFT) [4]. The global symmetry of the WCFT is $SL(2, R) \times U(1)$ and its local enhancement is the Virasoro–Kac–Moody algebra. The enhanced local symmetries have a clear dual interpretation from the gravity side in the context of AdS$_3$/CFT$_2$ correspondence. They reveal the enhancement of the asymptotic symmetry from the isometry of the AdS spacetime. More precisely, for AdS$_3$ gravity, the asymptotic symmetry group under Brown–Henneaux boundary conditions contains two copies of Virasoro symmetry [5]. Meanwhile, for the waped AdS$_3$ case [6,7], the algebra of asymptotic symmetries is isomorphic to the semi-direct product of a Virasoro algebra and an algebra of currents under Compère-Detournay boundary condition [8] (The local symmetry of WCFT can be also realized in AdS$_3$ gravity under the Compère–Song–Strominger boundary conditions [9]).

Recently, the enhanced symmetry was revealed for 2D Galilean field theories with anisotropic scaling symmetry [10]. The 2D Galilean field theories with global translations and anisotropic scaling symmetries are shown to have enhanced local symmetries which are generated by the infinite-dimensional spin-$k$ Galilean algebra. For 2D Galilean field

theories with isotropic scaling symmetry, the dual gravity theory is proposed to be three-dimensional and asymptotically flat [11,12] where the asymptotic symmetry is enhanced from the global 3D Poincare group to the infinite-dimensional Bondi–Metzner–Sachs (BMS) group. For the anisotropic case, the dual gravity theory is proposed [13] to be higher-dimensional Schrödinger geometry [14]. But the asymptotic symmetry of these geometries has not been addressed. Whether it can recover the enhanced local symmetry from the field theory side is not yet known.

The main purpose of this work is to find the gravitational duality for the enhancement of symmetry in [10]. We find a new 4D metric with the isometry group isomorphic to the global symmetry of the 2D Galilean field theories with anisotropic scaling, which presents a different realization of gravity dual for the 2D theory from the one in [13,14]. This metric describes a homogeneous spacetime with a constant curvature tensor. The 4D spacetime is supposed to be dual to the vacuum of the field theory. We show that Einstein gravity with quadratic-curvature extension admits the 4D spacetime as a vacuum solution when the coupling constants of the higher-derivative terms are specially adapted to the dynamical exponent $z$. The 4D spacetime, with the restricted range of the dynamical exponent $z$, can also be obtained from the Einstein–Proca theory. Then, we find a solution phase space of the higher-derivative theory, which admits the vacuum metric and yields precisely the infinite-dimensional spin-$k$ Galilean algebra in [10] as asymptotic symmetry. Since the duality is between 4D and 2D, we adopt a double expansion of inverse spatial directions. Correspondingly, the 2D Galilean theory is defined on the corner of the 4D spacetime boundary.

The organization of this paper is as follows. In Section 2, we present the vacuum solution and show that it has a constant curvature tensor. In Section 3, we comment on the gravitational theory that admits the vacuum solution. In Section 4, we show a solution phase space and derive the asymptotic symmetry of the phase space, namely the most generic residual gauge transformations that preserve the solution phase space. In Section 5, we derive the asymptotic symmetry algebra. We conclude in the last section.

## 2. Spacetime with Global Anisotropic Scaling Symmetry

The global symmetry of 2D Galilean field theory with anisotropic scaling consists of translations along two directions: the Galilean boost

$$x \rightarrow x - vt \,, \tag{1}$$

and the dilations (we set the parameter $c$ in [10] to be 1. This can always be realized by the rescaling $\lambda^c \rightarrow \tilde{\lambda}$, hence $\frac{d}{c} \rightarrow k$).

$$t \rightarrow \lambda t, \quad x \rightarrow \lambda^k x \,, \tag{2}$$

In Ref. [10], it is shown that the symmetry of the 2D Galilean field theory with the above global symmetry is enhanced to an infinite-dimensional spin-$k$ Galilean algebra. In plane modes, the algebra is given by

$$
\begin{aligned}
[l_n, l_m] &= (n - m)l_{n+m} \,, \\
[l_n, m_m] &= (kn - m)m_{n+m} \,, \\
[m_n, m_m] &= 0 \,.
\end{aligned} \tag{3}
$$

This algebra with $k = 1$ is precisely the BMS$_3$ algebra derived in [15,16].

From the holographic principle, the field theory is defined on the boundary of the dual gravity theory. The global symmetry of the field theory is the isometry of the bulk spacetime. For the 2D Galilean field theory with anisotropic scaling, the dual gravity

theory is four-dimensional. The 4D metric which we find with global anisotropic scaling isometry is

$$ds^2 = \frac{\ell^2 dr^2}{r^2} + r^{2z}dy^2 + r^2(2dxdt - ydt^2). \tag{4}$$

The metric is invariant under the translations along $t$ and $x$ directions plus the following global transformations,

Galilean embedding : $\quad x \to x - vt, \quad t \to t, \quad y \to y - 2v,$

scaling embedding : $\quad x \to \lambda^k x, \quad t \to \lambda t, \quad y \to \lambda^{k-1}y, \quad r \to \lambda^{-\frac{k+1}{2}}r, \tag{5}$

with

$$k = \frac{2+z}{2-z}. \tag{6}$$

The spacetime described by the metric (4) is homogeneous and has a constant curvature tensor. For constant $y$, the metric is the AdS$_3$ in planar coordinates with a 2D Minkowski boundary. The introduction of $y$ serves the purpose of breaking the 2D conformal group to Gallilean symmetry with anistropic scaling. Our construction of the background metric (4) is very different from the proposal in [13], where an extra null Killing direction associated with the coordinate $\xi$ was introduced. In our case, however, $y$ is not a Killing direction, but it is analogous to the radial coordinate $r$, which allows us to take an additional $y \to \infty$ limit in the asymptotic expansion. This makes the analysis of the asymptotic symmetry simpler and well controlled by both the $(r, y)$ coordinates. We shall return to this in the next sections.

The constant curvature tensor can be easily obtained from the vielbein formalism. A natural vielbein choice that respects the global isometry is

$$e^0 = \frac{\ell dr}{r}, \qquad e^1 = r^z dy, \qquad e^+ = r^{1-\frac{1}{2}z}dt, \qquad e^- = r^{1+\frac{1}{2}z}(dx - \tfrac{1}{2}ydt), \tag{7}$$

such that $ds^2 = e^0 e^0 + e^1 e^1 + 2e^+ e^-$. Thus, we have

$$de^0 = 0, \qquad de^1 = \frac{z}{\ell}e^0 \wedge e^1, \qquad de^+ = \frac{1 - \frac{1}{2}z}{\ell}e^0 \wedge e^+,$$

$$de^- = \frac{1 + \frac{1}{2}z}{\ell}e^0 \wedge e^- - \tfrac{1}{2}e^1 \wedge e^+. \tag{8}$$

The spin connections are

$$\omega^{\pm 0} = \frac{e^\pm}{\ell}, \qquad \omega^{+-} = \frac{z}{2\ell}e^0, \qquad \omega^{10} = \frac{ze^1}{\ell}, \qquad \omega^{1-} = \tfrac{1}{2}e^+. \tag{9}$$

We thus have the curvature tensor 2-form $\Theta^a{}_b = \frac{1}{2}R^a{}_{bcd}e^c \wedge e^d$ as

$$\Theta^+{}_+ = -\frac{1}{\ell^2}e^+ \wedge e^-, \quad \Theta^+{}_0 = \frac{1}{\ell^2}e^0 \wedge e^+, \quad \Theta^-{}_0 = \frac{1}{\ell^2}e^0 \wedge e^- + \frac{1-z}{2\ell}e^+ \wedge e^1,$$

$$\Theta^+{}_1 = -\frac{z}{\ell^2}e^+ \wedge e^1, \quad \Theta^-{}_1 = \frac{z-1}{2\ell}e^0 \wedge e^+ - \frac{z}{\ell^2}e^- \wedge e^1, \quad \Theta^0{}_1 = -\frac{z^2}{\ell^2}e^0 \wedge e^1. \tag{10}$$

Hence, the independent non-vanishing components of the Riemann tensor are

$$R^+{}_{++-} = -\frac{1}{\ell^2}, \qquad R^+{}_{0+0} = -\frac{1}{\ell^2}, \qquad R^-{}_{0+1} = \frac{1-z}{2\ell},$$

$$R^+{}_{1+1} = -\frac{z}{\ell^2}, \qquad R^0{}_{101} = -\frac{z^2}{\ell^2}. \tag{11}$$

The Ricci tensor is given by

$$R_{+-} = -\frac{z+2}{\ell^2}, \qquad R_{00} = -\frac{z^2+2}{\ell^2}, \qquad R_{11} = -\frac{z(z+2)}{\ell^2}. \tag{12}$$

and the Ricci scalar is

$$R = -\frac{2(z^2+2z+3)}{\ell^2}. \tag{13}$$

When $z = 1$, the metric (4) is just the 4D AdS spacetime.

As one can see from (6) that $k = -1$ requires $z \to \pm\infty$. So, the metric is not well defined for this particular choice. To include this limiting case, we make a coordinate transformation and redefine the parameter as follows,

$$r = \tilde{r}^{\frac{1}{z}}, \qquad \ell = z\tilde{\ell}. \tag{14}$$

The new metric admits a $z \to \pm\infty$ limit, and we obtain

$$ds^2 = \frac{\tilde{\ell}^2 d\tilde{r}^2}{\tilde{r}^2} + \tilde{r}^2 dy^2 - y dt^2 + 2dt dx. \tag{15}$$

The vielbein, spin connections and curvature can be simply obtained from the same treatment when taking the limit $z \to \pm\infty$.

## 3. Dual Gravity Theories

Einstein gravity with the most general quadratic-curvature extension in four dimensions is

$$\mathcal{L} = \sqrt{-g}(R - 2\Lambda_0 + \alpha R^2 + \beta R^{\mu\nu} R_{\mu\nu}). \tag{16}$$

Since the Gauss–Bonnet term is a total derivative, we do not add the Riemann squared term. The metric (4) is a solution of the theory (16) when the coupling constants $\alpha$, $\beta$ and the cosmological constant $\Lambda_0$ are specially chosen with respect to the dynamical exponent $z$,

$$z = 1: \qquad \Lambda_0 = -\frac{3}{\ell^2}, \qquad \text{no constraint on } \alpha \text{ and } \beta, \tag{17}$$

$$z \neq 1: \qquad \Lambda_0 = -\frac{(z^2+2z+3)}{2\ell^2}, \qquad 4\alpha(z^2+2z+3) + 2\beta(z^2+2) = \ell^2. \tag{18}$$

Note that when setting $\beta = 0$, the generic theory (16) is reduced to the ghost-free theory $\mathcal{L} = \sqrt{-g}(R - 2\Lambda_0 + \alpha R^2)$, which still admits the vacuum solution (4), and the coupling constant is completely fixed by

$$\alpha = \frac{\ell^2}{4(z^2+2z+3)}. \tag{19}$$

It is also of interest to examine whether our metric (4) can arise from Einstein theory with minimally coupled matter. The curvature tensor in the vielbein base implies that

$$G^{00} = \frac{1+2z}{\ell^2}, \qquad G^{11} = \frac{3}{\ell^2}, \qquad G^{+-} = \frac{z^2+z+1}{\ell^2}. \tag{20}$$

If the solutions are constructed from the Einstein theory with minimally coupled matter fields, then we can deduce that $T_{\text{tot}}^{ab} = G^{ab}$. In the diagonal base, we have

$$\rho = -\frac{z^2+z+1}{\ell^2} = -p_1, \qquad p_2 = \frac{3}{\ell^2}, \qquad p_3 = \frac{1+2z}{\ell^2}. \tag{21}$$

The null energy conditions (NEC) $\rho + p_i \geq 0$ impose the following conditions

$$2 - z - z^2 \geq 0 , \qquad z(1 - z) \geq 0 . \tag{22}$$

This requires that $0 \leq z \leq 1$. As a concrete example, we consider Einstein gravity coupled to massive vector theory (Proca theory)

$$\mathcal{L} = \sqrt{-g}(R - 2\Lambda - \tfrac{1}{4}F^2 - \tfrac{1}{2}\mu^2 A^2) . \tag{23}$$

To admit metric (4) as a solution, the cosmological constant $\Lambda$, the coupling constant $\mu$, and the vector field $A$ in the Proca theory should have the following forms,

$$\Lambda = -\frac{z^2 + z + 4}{2\ell^2} , \qquad \mu^2 = \frac{2z}{\ell^2} , \qquad A = \sqrt{\frac{2(1 - z)}{z}} \, r^z \, dy . \tag{24}$$

Thus, the reality condition requires that $0 < z \leq 1$.

In fact, although the total energy momentum tensor in the Einstein–Proca–$\Lambda$ theory satisfies only the NEC but violates both the weak and strong energy condition, the energy-momentum tensor of the Proca field

$$\rho^A = \frac{(1 - z)(2 + z)}{\ell^2} = -p_1^A = p_3^A , \qquad p_2^A = -\frac{(1 - z)(2 - z)}{2\ell^2} , \tag{25}$$

satisfies all the energy conditions, namely the strong energy condition and dominant energy condition, since $0 < z \leq 1$. It is well known that the culprit of violating weak or strong energy condition in these cases is the cosmological constant.

## 4. Asymptotic Symmetries

The complete set of gauge transformation of Einstein gravity with quadratic-curvature extension is generated by infinitesimal diffeomorphism (since the Proca theory admits metric (4) as a solution only when $0 < z \leq 1$, we consider Einstein gravity with quadratic-curvature extension for deriving the asymptotic symmetry to have a consistent study for a generic value of $z$),

$$\delta_{\xi} g_{\mu\nu} = \mathcal{L}_{\xi} g_{\mu\nu} . \tag{26}$$

The asymptotic symmetry is the residual gauge transformation that preserves the required gauge and boundary conditions. We will follow the Fefferman–Graham gauge,

$$g_{rr} = \frac{\ell^2}{r^2} , \qquad g_{rA} = 0 , \tag{27}$$

where $A = (y, t, x)$. The residual gauge transformation preserving the Fefferman–Graham gauge can be solved as follows:

- $\mathcal{L}_{\xi} g_{rr} = 0 \implies \xi^r = -\tfrac{1}{2} r \Psi(y, t, x).$
- $\mathcal{L}_{\xi} g_{rA} = 0 \implies \xi^A = Y^A(y, t, x) - \frac{\ell^2}{2} \partial_B \Psi \int_r^{\infty} \frac{dr'}{r'} g^{AB},$

where $g^{AB}$ is the inverse metric. For a generic $z$, it is very hard to impose boundary conditions to study asymptotic symmetries in a unified way. Alternatively, we apply the solution phase space method [17–19] to investigate the asymptotic (symplectic) symmetry of the system. The solution phase space method was originally introduced for the Near-Horizon Extremal Geometries (NHEGs) [17,18]. Since the NHEG is absent of dynamical physical perturbations [20], it is natural to consider the action of diffeomorphisms on the NHEG to construct the classical phase space. In our study, the theory (16) depends on the dynamical exponent $z$, namely for each choice of $z$, there is a corresponding theory and fall-off conditions. So, the solution phase space method is particularly useful for us to study asymptotic symmetries for a generic choice of $z$.

We find a solution phase space of the theory (16), which is given by

$$ds^2 \;=\; \frac{\ell^2}{r^2}dr^2 \;+\; r^{2z}\Phi(t)^2dy^2 \;+\; 2r^2 \quad dtdx \;-\; r^2\left[yf_1(t)+f_2(t)+xf_3(t)+\frac{\ell^2 f_3(t)\Phi'(t)}{2\Phi(t)r^2}-\frac{\ell^2\Phi''(t)}{\Phi(t)r^2}\right]dt^2, \quad (28)$$

where a prime denotes a derivative on $t$. There are four arbitrary functions of time $t$, namely, $\Phi(t), f_1(t), f_2(t), f_3(t)$ which represent four types of independent diffeomorphisms. They are dynamical fields of the phase space. However, they only represent boundary dynamics as they are introduced from the action of diffeomorphisms on the homogeneous spacetime (4). There is no propagating degree of freedom in the phase space. The dual theory of the 2D Galilean field theory is from boundary gravity in the context of AdS/CFT [5,21].

The most generic residual gauge transformation preserving the phase space is characterized by

$$\begin{aligned}
\Psi(y,t,x) &= \psi(t)\,, & Y^y(y,t,x) &= Y_1 y + Y_2\,, \\
Y^t(y,t,x) &= L(t)\,, & Y^x(y,t,x) &= [\psi(t)-\partial_t L(t)]x + M(t)\,,
\end{aligned} \quad (29)$$

where $Y_1$ and $Y_2$ are real and constant. Note that we choose all symmetry parameters to be field independent.

To manifest the fact that the spacetime is scaling invariant asymptotically, we need to perform a double expansion in terms of both $\frac{1}{r}$ and $\frac{1}{y}$ in the region of a large $r$ and large $y$ and set $f_1(t) = \Phi(t)$. The asymptotic form of the metric is

$$ds^2 = \frac{\ell^2}{r^2}dr^2 + r^{2z}\Phi(t)^2dy^2 + 2r^2dtdx - r^2y\Phi(t)dt^2 + \text{sub-leading terms}\,, \quad (30)$$

The leading part of the metric is invariant under the scaling transformation,

$$\begin{aligned}
x &\to \lambda^k x\,, & t &\to \lambda t\,, & r &\to \lambda^{-\frac{k+1}{2}}r\,, \\
y &\to \lambda^{k-1-a}y\,, & \Phi(t) &\to \lambda^a\Phi(t)\,.
\end{aligned} \quad (31)$$

The condition $f_1(t) = \Phi(t)$ will further yield that

$$\psi(t) = (k+1)L'(t)\,. \quad (32)$$

To summarize, the asymptotic symmetry that preserves the scaling-invariant phase space (28) is generated by

$$\begin{aligned}
\xi^r &= -\frac{1}{2}(k+1)L'(t)r\,, \\
\xi^y &= Y_1 y + Y_2\,, \\
\xi^t &= L(t)\,, \\
\xi^x &= M(t) + kxL'(t) - \frac{\ell^2}{4}(k+1)\frac{L''(t)}{r^2}\,.
\end{aligned} \quad (33)$$

## 5. Asymptotic Symmetry Algebra

The asymptotic Killing vectors (33) satisfy the standard Lie algebra

$$[\xi_1,\xi_2]^\mu = \xi_1^\nu\partial_\nu\xi_2^\mu - \xi_2^\nu\partial_\nu\xi_1^\mu\,. \quad (34)$$

The algebra is closed and the Jacobi identity of the symmetry algebra is guaranteed by the Jacobi identity of the Lie algebra of three vectors. In terms of the basis vectors

$$L_m = \frac{1}{2}(k+1)(m+1)t^m r\partial_r - t^{m+1}\partial_t - k(m+1)t^m x\partial_x$$
$$+ \frac{\ell^2}{4}(k+1)m(m+1)t^{m-1}\frac{1}{r^2}\partial_x \,, \tag{35}$$
$$M_m = -t^{m+k}\partial_x \,, \qquad B = \partial_y \,, \qquad D = y\partial_y \,,$$

the asymptotic algebra is

$$[L_n, L_m] = (n-m)L_{m+n} \,, \tag{36}$$
$$[L_n, M_m] = (kn-m)M_{m+n} \,, \tag{37}$$
$$[M_n, M_m] = 0 \,, \tag{38}$$
$$[B,B] = 0 \,, \quad [D,D] = 0 \,, \quad [D,B] = -\frac{k-1}{k+1}B \,, \tag{39}$$
$$[B, L_m] = 0 \,, \qquad [B, M_m] = 0, \tag{40}$$
$$[D, L_m] = 0 \,, \qquad [D, M_m] = 0 \,. \tag{41}$$

The first three lines of the above equations are precisely the symmetry algebra (3) derived from the dual 2D field theory side [10]. We have two more generators $D$ and $B$ from the extra dimension $y$ which commute all the modes $L_m$ and $M_m$.

The Killing vector that generates the Galilean transformation of the background metric (4) is

$$G \equiv \partial_y + \frac{1}{2}t\partial_x \,. \tag{42}$$

It is of course included in the asymptotic Killing vectors which in mode expansion (35) is given by

$$G = B - \frac{1}{2}M_{1-k} \,. \tag{43}$$

The scaling symmetry

$$\Delta \equiv -\frac{1}{2}r\partial_r + \frac{k-1}{k+1}y\partial_y + \frac{t}{k+1}\partial_t + \frac{kx}{k+1}\partial_x \,, \tag{44}$$

in mode expansion (35) is given by

$$\Delta = \frac{k-1}{k+1}D - \frac{1}{k+1}L_0. \tag{45}$$

Two translations along the $t$ and $x$ directions are $L_{-1}$ and $M_{-k}$. The four Killing vectors form a closed subalgebra of the asymptotic symmetry algebra,

$$[G,G] = 0 \,, \quad [\Delta, \Delta] = 0 \,, \quad [\Delta, G] = -\frac{k-1}{k+1}G \,, \tag{46}$$

$$[G, L_{-1}] = -\frac{1}{2}M_{-k} \,, \qquad [\Delta, L_{-1}] = -\frac{1}{k+1}L_{-1} \,, \tag{47}$$

$$[G, M_{-k}] = 0 \,, \qquad [\Delta, M_{-k}] = -\frac{k}{k+1}M_{-k} \,. \tag{48}$$

The commutators of $G$ and $\Delta$ with other modes are

$$[G, L_m] = \frac{1}{2}[(m+1)k - 1]M_{m+1-k},\tag{49}$$

$$[\Delta, L_m] = \frac{m}{k+1}L_m,\tag{50}$$

$$[G, M_m] = 0,\tag{51}$$

$$[\Delta, M_m] = \frac{m}{k+1}M_m.\tag{52}$$

## 6. Conclusions

In this paper, we present an alternative realization of the holographic dual for the 2D Galilean field theory with anisotropic scaling. The isometry of the bulk spacetime is isomorphic to the global symmetry of the 2D Galilean field theory. And the bulk spacetime is a vacuum solution of the Einstein gravity with quadratic-curvature extension. For some restricted range of the dynamical exponent $z$, the bulk spacetime is also a solution of the Einstein–Proca theory. We find a solution phase space which admits the vacuum solution. The residual gauge transformation preserving the solution phase space, namely the asymptotic symmetry of this system, recovers precisely the enhanced local symmetry of the 2D Galilean field theory in [10]. Since the dual gravity theory is in 4D, our proposal is an example of a codimension-2 holography, and we have two more symmetry transformations from the extra dimensions. We show that the algebra of the two extra symmetry generators is closed and the generators of the two extra symmetry commute with all the generators of the 2D enhanced local symmetry. In our construction, the extra dimension is not a Killing direction and there is no conserved quantity associated to this extra direction. So, it can not be interpreted as a particle-number circle, such as in the Schrodinger theories [22] arisen from the TsT transformation [23] or Melvin twist of AdS space [24]. The extra dimension $y$ in our case is not compacted, so one cannot get rid of it by dimensional reduction [25]. However, in our construction, one can realize the enhanced local symmetry of the boundary field theory as asymptotic symmetry of the dual gravity system in a double expansion in large $r$ and large $y$. Correspondingly, the dual field theory is supposed to be defined on the corner of the 4D boundary; see, e.g., in [26] for a comprehensive introduction of the corner proposal and references therein.

As an ending remark, it is worth mentioning that our construction is temporarily restricted in 4D with a ground state. It is definitely of interest to generalize our construction to higher dimension or black hole solutions (thermal state) to incorporate various anisotropic invariant field theories in different perspectives, such as the theories studied in [27–32].

**Author Contributions:** Conceptualization, H.L., P.M. and J.W.; methodology, H.L., P.M. and J.W.; validation, H.L., P.M. and J.W.; formal analysis, H.L., P.M. and J.W.; investigation, H.L., P.M. and J.W.; writing—original draft preparation, P.M.; writing—review and editing, H.L., P.M. and J.W. All authors have read and agreed to the published version of the manuscript.

**Funding:** This research was funded by the National Natural Science Foundation of China under Grants No. 11905156, No. 11875200, No. 11975164, No. 11935009, No. 11947301, No. 12047502 and by the Natural Science Foundation of Tianjin under Grant No. 20JCYBJC00910.

**Data Availability Statement:** Not applicable.

**Conflicts of Interest:** The authors declare no conflict of interest.

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
