# Peer review of "Anisotropic Scaling Non-Relativistic Holography: A Symmetry Perspective"

_symmetry, doi:10.3390/sym15081579_

Round 1

Reviewer 1 Report

The manuscript “Anisotropic scaling non-relativistic holography: a symmetry perspective” constructs the gravity dual of a 2-dimensional Galilean theory with anisotropic scaling, providing a match of asymptotic symmetries up to two additional generators in the bulk. The authors further provide two bulk theories that could yield the ground state for a theory with the given symmetries: a specific quadratic gravity system and an Einstein-Proca system. The analysis is correct and clearly explained, and to my knowledge, the solutions provided are new.

However, I have some minor points that should be addressed to improve the readability and overall quality of the paper and, therefore, I recommend a minor revision of the manuscript:

- Throughout this paper, the authors refer to the boundary theory as a “conformal field theory”. This is wrong and this terminology shouldn’t be used, as the symmetries of the theory do not contain the conformal group.

- The solutions provided are an example of a “codimension-2 holography”, since the bulk is 4-dimensional and the dual is 2-dimensional. Other similar examples are the Schrodinger theories that arise from TST transformations or Melvin twist (hep-th/0502086, hep-th/0306131) of AdS spaces (arXiv:1009.4997). In these examples, one obtains an extra dimension interpreted as the “particle-number circle”. What is the interpretation of the extra dimension in this model? Is it possible to do a dimensional reduction to get rid of such a direction (as in, e.g., arXiv:1007.2184)?

- Finally, the authors may add a discussion on possible higher dimensional and black hole (beyond the ground state) generalizations. There exist several anisotropic versions of Lifshitz/Schrodinger invariant theories, both in top-down holography (arXiv:1105.3472, 1106.1637) and bottom-up (1608.02970, 1708.05691, 1907.05744, 2011.09474).

Author Response

We thank the referee for the useful comments. We have revised our manuscript following the suggestions. Our responses are as follows:

1) We change the terminology for the boundary theory to 2D Galilean field theory.

2) We give some comments in the last section about the extra dimensions in the holography that we are studying.

3) We mention the interests of finding black hole solutions and higher dimensional extension of our proposal in the last section following the line of the references pointed out by the referee.

The changes that are relevant to the comments from the first referee in the revised manuscript have been highlighted in blue text.

Reviewer 2 Report

The authors present a metric in 4D whose asymptotic symmetries match those of a 2D Galilean CFT. However, there are many points throughout the paper that must be improved:

1. There is no explanation for why the metric given by eqn. 2.4 is an interesting metric to study. There is also no explanation of the physics behind this metric (there is no g_xx term, and the Ricci scalar is given by a 3D value---can you explain these features, for instance?)

2. The metric is then found from a higher-curvature action with specified coefficients. Have you checked that this gives a sensible solution, a.k.a. that the stress tensor as defined by higher-curvature corrections obeys the relevant energy conditions? The same question goes for the Proca theory.

3. Obtaining the metric 4.3 is not obvious to me, and there is no explanation of the phase space method, nor how the authors used it to obtain the given solution.

4. The symmetries may be okay, but this is only valid for one solution, that is, for one state. In order to claim a holographic duality, one must have a family of solutions corresponding to the CFT in question, and one must check other quantities in order to obtain a proper duality. 

Overall, the paper reads as a vague collection of calculations as there is little to no discussion of the physics behind them. Unless the authors clarify their work in great detail and add relevant background material, I cannot even assess if their calculations are correct or if the work is sensible.

English is mostly fine, but there are some typos. An example is the first sentence in the second paragraph of the first page.

Author Response

We thank the referee for the comments. We have revised our manuscript following part of the suggestions. Our responses are as follows:

1) The interest of the metric in eq. (2.4) is purely from the holographic side. Its isometry is isomorphic to the global symmetry of the 2D field theory that we are interested. Similar situations for constructing metric can be found from Ref. [13-14] of our manuscript.

2) It is not clear to us the meaning of asking the stress tensor defined by higher-curvature corrections. The energy conditions are only relevant to matter fields that are minimally coupled to gravity. For the higher derivative theory in our case, it is a purely gravitational theory without any matter field. It is thus a vacuum solution and hence all energy conditions are satisfied. It is perhaps sensible to ask, if the theory is Einstein and the metrics are supported by minimally coupled matter, what kind of constraint on z if we insist on appropriate energy conditions. This issue has been addressed in the revised manuscript in section 3. The changes that are relevant to the comments from the second referee in the revised manuscript have been highlighted in red text.

3) We give some generic descriptions about the phase space method.

4) We agree with the referee that one must check several quantities in order to obtain a proper duality. Symmetry aspect is one of those important quantities that one should check for proposing a holographic duality. This is precisely what we have done in this work. Our point is that asymptotic symmetry is a self-contained perspective for studying a holographic duality, see, e.g., from Refs. [5,8,9,17-20]. Asymptotic symmetry has never been investigated for the case of our interests as confirmed by another referee. We would like to leave other quantities, such as correlation functions, for the future investigations.

5) We think the statement that our work is only a vague collection of calculations is not quite a fair assessment of our paper. The embedding of Galilean symmetry with anisotropic scaling into curved spacetime as geometric symmetry is not straightforward. Our homogeneous metric is a nontrivial construction and can arise in reputed gravity theories. It is certainly true that the metric and its asymptotic symmetry raise many valid questions that we are not able to answer at this stage. Our work may not meet the criteria of publishing in this journal. However, the topic of holographic nature of Galilean symmetry is not a mature subject and there is no readily available technique to follow. Our work provides an early effort of exploring the subject.

6) We have corrected a few typos.

Round 2

Reviewer 2 Report

The comment about higher curvature corrections is a simple one: the additional higher-curvature terms can be "moved to the RHS" of Einstein's equations, providing a stress tensor representation. 

Regardless, the authors have sufficiently addressed the other points raised, and I appreciate the first efforts in trying to understand this system better. Therefore, I recommend the paper be published.